# Loss of Argininosuccinate Synthetase-1 (ASS1) Occurs in Esophageal Adenocarcinoma and Represents a Promising Biomarker for Therapy with Pegargiminase

**DOI:** 10.3390/cancers17223624

**Published:** 2025-11-11

**Authors:** Karl Knipper, Su Ir Lyu, Eleni Tzitzili, Sarah-Michele Spielmann, Christiane J. Bruns, Thomas Schmidt, Felix C. Popp, Alexander Quaas

**Affiliations:** 1Department of General, Visceral, Thoracic and Transplantation Surgery, Faculty of Medicine and University Hospital of Cologne, University of Cologne, 50937 Cologne, Germany; 2Institute of Pathology, Faculty of Medicine and University Hospital of Cologne, University of Cologne, 50937 Cologne, Germany

**Keywords:** esophageal adenocarcinoma, Argininosuccinate Synthetase 1, ASS1, pegargiminase, personalized medicine, biomarker

## Abstract

Downregulation of Argininosuccinate Synthetase-1 (ASS1) supports tumor progression. Following the downregulation of ASS1, tumor cells rely on an external arginine delivery–a potential “Achilles heel” of these tumors. The treatment of patients with ASS1-deficient malignant pleural mesothelioma with pegargiminase, an arginine deprivation therapy agent, has shown prolonged survival. We aimed to assess the frequency of ASS1 downregulation in patients with esophageal adenocarcinoma. In this study, we performed ASS1 immunohistochemical stainings in 97 tumors. Among the patients included in this analysis, 6.2% exhibited an ASS1 loss, and a further 6.2% showed low ASS1 expression. Our results indicate that a significant number of patients with esophageal adenocarcinoma (12.4%) could potentially be eligible for arginine deprivation therapy. These findings underscore the need for clinical trials investigating the efficacy of pegargiminase treatment in patients with esophageal adenocarcinoma.

## 1. Introduction

FDA-approved targeted therapy options for esophageal adenocarcinoma are steadily expanding. Recently, nivolumab, an antibody against the programmed cell death ligand-1 receptor, proved to prolong disease-free survival as adjuvant therapy in patients with esophageal or gastroesophageal junction cancer [1]. However, despite curative treatment with neoadjuvant chemoradiotherapy, surgical resection, and adjuvant nivolumab, the disease-free survival remains limited to 22.4 months [1]. This underlines the pressing need for additional targeted therapy options for patients with esophageal adenocarcinoma. Furthermore, personalized cancer treatment options need to be expanded to improve treatment efficacy. For instance, patients with HER2-positive advanced gastric and gastroesophageal junction carcinoma receiving targeted therapy with trastuzumab, an antibody targeting HER2, demonstrated an improvement in disease-free survival [2]. Although the results were modest, extending survival by only 1.5 months, they underline the importance of identifying new molecular targets that can shape personalized treatment schemes.

Argininosuccinate synthetase-1 (ASS1) catalyzes the formation of argininosuccinate from citrulline and aspartate in the urea cycle [3]. This process is fundamental for the production of the non-essential amino acid arginine [4]. The suppression of ASS1 has been shown to enhance migration and invasion of hepatocellular carcinoma cells in vitro, while ASS1 overexpression inhibits metastasis in vivo [5]. As a result, *ASS1* is widely regarded as a tumor suppressor gene [6,7]. Downregulation of ASS1 is correlated with worse patient survival in various cancers, including non-small-cell lung cancer and nasopharyngeal carcinoma [8,9].

Cancer cells, being highly metabolically active, rely on a sufficient supply of essential components for growth and proliferation [10]. Interestingly, ASS1 is downregulated in certain tumor entities [8,9]. The key mechanism proposed in ASS1 downregulation is the hypermethylation of the *ASS1* promoter, resulting in loss of ASS1 expression in cancer cells [6]. For some patients, the downregulation causes an auxotrophy, presenting an “Achilles heel” in these tumors. The novel therapeutic agent pegargiminase exploits this vulnerability. The pegylated arginine deiminase degrades arginine, creating arginine deficiency in ASS1-deficient cells and thereby inhibiting cancer cell growth, as demonstrated in melanoma and hepatocellular carcinoma cells in vitro [11]. In clinical settings, treatment with pegargiminase has been evaluated in patients with ASS1-deficient malignant pleural mesothelioma. In this cohort, patients receiving pegargiminase showed significantly prolonged disease-free survival compared to those receiving best supportive care. Furthermore, stable disease was observed significantly more frequently in the pegargiminase group [12].

In this study, we aimed to determine whether ASS1-loss occurs in esophageal adenocarcinoma and to evaluate the prognostic significance of ASS1 expression in this cancer type.

## 2. Materials and Methods

### 2.1. Patients and Tumor Samples

Inclusion criteria for this project were the diagnosis of esophageal adenocarcinoma and a conducted tumor resection with curative intention. We excluded patients from survival analyses if survival data were missing or the survival period was less than 30 days. All patients underwent resection at the University Hospital of Cologne between 2007 and 2024. Written informed consent for participation in the tissue and data bank was obtained from each patient. The study was conducted following the Declaration of Helsinki.

Furthermore, this study was approved by the Ethics Committee of the University Hospital of Cologne (Ethics Committee number: 21-1146). The clinical data were collected in our data bank prospectively and analyzed retrospectively. Overall survival was defined as the period between the date of operation and death or loss of follow-up. For survival analysis, only patients with a follow-up of at least 30 days were included (*n* = 67). The disease stage was evaluated following the 7th edition of the Union for International Cancer Control [13]. Minor response was defined as grade I and II of the Cologne regression grade system, which translates to ≥10% vital residual tumor cells [14].

### 2.2. Immunohistochemistry

For immunohistochemical staining, the automatic staining system Leica BOND-MAX with the Leica Bond Polymer Refine Detection Kit (Leica Biosystems, Wetzlar, Germany) was used. The following antibodies against ASS1 were used: D4O4B, Cell Signaling Technology, Danvers, MA, USA; EPR12398, Abcam, Cambridge, UK; HPA020896, Sigma-Aldrich, St. Louis, MO, USA. Dilutions, reagents, and control tissues were conducted following the manufacturers’ recommendations. 4 µm thick slices of paraffin-embedded sections were used. A tissue microarray with 50 patients was stained for the comparison of the three antibodies with the RNAScope results. The final ASS1 stainings were conducted on whole tumor sections. The samples were screened for endothelial cells as an internal positive control of the stainings performed.

The staining results were independently analyzed by two experienced pathologists. ASS1 expression was categorized as either positive (>10% positive cancer cells), low positive (≤10% and >0% positive cancer cells), or negative (0% positive cancer cells).

### 2.3. RNAScope^TM^

The RNAScope™ stainings were conducted on 5 µm thick paraffin-embedded whole tumor slides. The assay was performed as described before [15,16]. In short, the sections were deparaffinized, pretreated for 30 min, digested and hybridized with the RNAScope™ probe-Hs-ASS1 (431291, Advanced Cell Diagnostics, Newark, NJ, USA) at 40 °C, and counterstained with hematoxylin for 10 s. Then, the signal was developed using the RNAScope™ 2.5 HD Assay-RED (322360, Advanced Cell Diagnostics, Newark, NJ, USA). The results were evaluated by two experienced pathologists following the manufacturers’ guidelines: score 0 = no or less than one molecule per 10 cells; score 1 = 1–3 dots/cell; score 2 = 4–9 dots/cell; score 3 = 10–15 dots/cell; score 4 = >15 dots/cell. Stainings were considered positive if a score higher than 0 was achieved.

### 2.4. Statistical Analysis

Statistical analyses were conducted using IBM SPSS Statistics (Version 29.0.1.1, IBM, Armonk, NY, USA). Kaplan–Meier curves and log-rank tests were performed for survival analyses. Qualitative variables were compared with the chi-square test. Interdependencies between survival and clinicopathological variables were analyzed with univariable and multivariable Cox regression analyses. Only variables that showed *p*-values below 0.2 in univariable Cox regression analyses were studied in multivariable Cox regression models. *p*-values below 0.05 were considered statistically significant.

## 3. Results

To identify the most reliable antibody for ASS1 immunohistochemical staining, we compared the staining results of three different antibodies with corresponding RNAScope in situ analyses in a tissue microarray comprising fifty esophageal adenocarcinoma samples (Appendix A). Among the tested antibodies, D4O4B, produced by Cell Signaling Technology, demonstrated the closest correlation with the RNAScope results and was therefore selected for further analysis.

Using the Cell Signaling antibody, we subsequently performed ASS1 immunohistochemical staining on whole tumor sections from 97 patients (and corresponding metastases in ASS1-loss tumors). The included patients had been diagnosed with esophageal adenocarcinoma and underwent curative-intent surgery between 2007 and 2024 at the University Hospital of Cologne. Patient characteristics are depicted in Table 1. Almost all patients suffered either from adenocarcinoma of the gastroesophageal junction (AEG) type I (45.4%) or from AEG type II (52.6%). Among the cohort, fifty patients (51.5%) received neoadjuvant therapy. Only 36.2% (*n* = 17) of the patients who underwent primary surgery were diagnosed with stage I disease initially. 30 patients (63.8%) either received a recommendation for primary surgery from our interdisciplinary tumor board conference due to multimorbidity or advanced age, or declined perioperative therapy and opted for primary resection. The majority of our cohort received an Ivor Lewis esophagectomy (85.6%), whereas fourteen patients (14.4%) underwent a transhiatal extended gastrectomy. All patients received a two-field lymphadenectomy. Of the included patients, 24 (24.7%) were operated openly, 54 (55.7%) were operated via hybrid technique, and 19 (19.6%) received a total minimally invasive resection. Postoperative complications occurred in 61 patients (62.9%). Severe complications (Clavien-Dindo classification ≥ 3) were recorded in 27 patients (27.8%). An anastomotic leakage occurred in 9 patients (9.3%).

The immunohistochemical staining results were categorized into three groups: ASS1 loss (absence of ASS1 protein expression in tumor cells), low ASS1 expression (less than 10% of tumor cells with ASS1 expression), and positive ASS1 expression (more than 10% of tumor cells ASS1 positive) (Figure 1A–C). The majority of patients exhibited positive ASS1 expression (87.6%). Six patients (6.2%) demonstrated low ASS1 expression, and six patients (6.2%) showed a complete ASS1 loss. Additionally, we examined thirteen normal mucous membranes for their ASS1 expression (squamous epithelial esophageal mucosa: *n* = 10, Barrett’s mucosa: *n* = 3). ASS1 is expressed in the basal keratinocytes of normal squamous epithelial esophageal mucosa. The remaining maturing squamous epithelia are ASS1-negative. The foveolar epithelia of the Barrett’s mucosa show a low ASS1 expression.

General patient characteristics were compared between these three patient groups (Table 1). Here, patients with ASS1 loss or low ASS1 expression were significantly younger than patients with positive ASS1 expression (*p* = 0.007). Furthermore, all patients with ASS1 loss who underwent neoadjuvant therapy showed a minor response to the administered neoadjuvant therapy. Although not statistically significant, a trend was observed indicating that patients with ASS1 loss or low ASS1 expression were more likely to have received neoadjuvant or perioperative therapy (*p* = 0.063). A higher Clavien-Dindo classification was associated with positive ASS1 expression (*p* = 0.018).

To evaluate the homogeneity of ASS1 loss, we screened the whole tumor sections when ASS1 loss was confirmed. The results revealed a homogeneous loss of ASS1 staining throughout the tumors. Additionally, we stained the corresponding lymph node metastases in these six patients and one corresponding brain metastasis. Consistently, the ASS1 loss was homogeneously detectable in all metastatic sites.

Next, we performed survival analyses to investigate the impact of ASS1 loss on patients’ overall survival. No significant difference in overall survival was observed in patients with ASS1 loss and low ASS1 expression compared to patients with positive ASS1 expression (*p* = 0.721, Figure 1D).

Furthermore, we conducted univariable Cox regression analyses to assess our data for confounding variables (Table 2). We included the following factors: sex, patient age, Clavien-Dindo classification, pathological tumor status, pathological lymph node status, lymph vessel infiltration, blood vessel infiltration, and ASS1 expression. High pathological tumor status ((y)pT) was significantly associated with worse patient survival compared to tumors with lower tumor status (HR = 3.334, 95% confidence interval = 1.165–9.543, *p* = 0.025).

A multivariable Cox regression analysis including all variables, which showed a *p*-value below 0.2 in our univariable Cox regression analyses was performed (Table 3). Patient age, Clavien-Dindo classification, pathological tumor status, pathological lymph node status, and blood vessel infiltration were included. High Clavien-Dindo classification (≥IIIb) and a high pathological tumor status (≥2) were identified as significant, independent risk factors for worse patient overall survival in our patient cohort (Clavien-Dindo classification: HR = 4.766, 95% confidence interval = 1.450–15.677, *p* = 0.010, (y)pT: HR = 4.542, 95% confidence interval = 1.363–15.132, *p* = 0.014).

In summary, 6.2% of the esophageal adenocarcinomas showed an ASS1 loss, and 6.2% showed a low ASS1 expression.

## 4. Discussion

This study showed that a complete loss of ASS1 expression in tumor cells could be observed in 6.2% of all patients with esophageal adenocarcinoma. In addition, a further 6.2% of tumors showed ASS1 expression in less than 10% of their tumor cells (ASS1 low). To our knowledge, we are the first to report the existence of ASS1-deficient esophageal adenocarcinomas in a cohort consisting solely of this entity.

Our findings indicate that a significant subgroup of patients with esophageal adenocarcinoma may be eligible for arginine deprivation therapy. Despite this, no clinical trial has yet evaluated pegargiminase in this patient population. The mentioned therapy option proved to offer a significantly increased progression-free survival as well as overall survival in patients with pleural mesothelioma in a phase 2-3 trial, if administered with chemotherapy [17].

Pegargiminase has also been evaluated in other tumor entities. A phase 1 study by Chang et al. involving patients with various solid tumors—including cholangiocarcinoma, esophageal cancer, and gastric cancer–reported a therapy response in 24% of all included patients [18]. A phase 1 trial in patients with non-small-cell lung cancer showed even more promising results with an achieved disease control in 85.7% of cases [19]. These notable differences in treatment response could be explained by several factors.

First, Chang et al. included patients with different solid cancer entities [18]. For example, information on the histological subtypes of esophageal carcinoma–adenocarcinoma or squamous cell carcinoma–of the seven included patients with this tumor entity was not provided by Chang et al. [18]. Different tumor types are likely to differ in their pathomechanisms and susceptibility to arginine deprivation therapy. Underlining this and in contrast with the general definition of *ASS1* as a tumor suppressor gene, according to some studies, the overexpression of ASS1 could lead to significantly more migration and proliferation in ovarian cancer cells in vitro [20]. Moreover, some tumor types may harbor therapy escape mechanisms due to distinct tumorigenesis processes.

Second, the eligibility criteria differed between studies. Chang et al. included patients regardless of their ASS1 status [18], while Szlosarek et al. included only those with ASS1-deficient tumors [19]. This difference likely contributed to the higher response rates observed in the ASS1-deficient cohort, further emphasizing the need for future clinical studies to establish patient selection criteria and focus on the populations most likely to benefit from arginine-deprivation therapy. Moreover, the existing trials used different cut-offs for defining ASS1 deficiency. Some trials defined ASS1 loss as less than 5% positive cancer cells in immunohistochemical staining [6,21], while others used a threshold of less than 50% positive cancer cells [12,22], or employed only semiquantitative evaluation systems [23]. In our study, we defined low ASS1 expression as less than 10% positive cancer cells and ASS1 loss as the complete absence of ASS1 expression in the tumor cells.

Another crucial factor affecting comparability of the various studies is the choice of the antibody. Studies to date have used a variety of antibodies, compromising the comparability and standardization. To address this, we evaluated three antibodies by comparing their staining patterns with RNAScope analyses and selected the antibody that provided the most correlating staining.

To ensure consistency and comparability in future research, it is essential to standardize cut-offs for ASS1 deficiency and the antibodies used for staining. Such standardization would enhance the reliability of data and facilitate cross-study comparisons.

In our study, we found no significant survival difference between patients with ASS1 loss and those with positive ASS1 expression in esophageal adenocarcinoma. Contrary to our findings, worse patient survival for patients with ASS1 loss could be observed in other entities, for example, gastric cancer, hepatocellular carcinoma, or myxofibrosarcoma [6,24,25]. However, Ding et al. could also not detect an impact on ASS1 expression and patient survival in esophageal carcinoma, similar to the results in our study [26]. The absence of a survival difference in both studies may be due to small cohort sizes. Once more, Ding et al. did not distinguish between esophageal adenocarcinoma and esophageal squamous cell carcinoma [26], thus limiting the interpretability of their findings regarding the rate of ASS1 loss and its implications for patient survival in patients with esophageal adenocarcinoma. Our Cox regression analyses revealed a high Clavien-Dindo classification (≥IIIb) and a high pathological tumor status (≥2) as independent risk factors for worse patient overall survival. The confirmation of these well-established risk factors in our patient cohort supports the representativeness of our study population.

Interestingly, we could describe that patients with ASS1 loss or low ASS1 expression were significantly younger compared to patients with positive ASS1 expression in our cohort (*p* = 0.007). Previous studies have suggested that esophageal adenocarcinomas may be biologically more aggressive in younger patients. In a retrospective cohort study of patients with esophageal adenocarcinoma, age younger than 45 years was associated with more advanced tumor and lymph node status. Furthermore, these patients showed a significantly worse disease-specific survival compared to older age groups [27]. On the other hand, ASS1 deficiency could be linked to higher cell proliferation rates in in vitro experiments [28,29], while an *ASS1* knockout in mouse embryonic stem cells led to decreased apoptosis [28]. We propose that ASS1 deficiency contributes to a more aggressive tumor biology, which may explain its higher prevalence in younger patients.

To evaluate the homogeneity of ASS1 loss within individual tumors, we conducted staining on whole tumor slides as well as on lymph node metastases and solid organ metastases, where available. Our results confirm that the ASS1 loss could be detected homogeneously in all examined samples. This consistency enhances the diagnostic reliability of ASS1 staining when performed on biopsies. This is particularly relevant in clinical settings, as all tumors with ASS1 loss following neoadjuvant therapy showed non-response to the administered treatment. Furthermore, the frequency of ASS1 loss or low ASS1 expression was higher in patients following neoadjuvant therapy (*p* = 0.063; barely not statistically significant). This could imply that ASS1 loss or low ASS1 expression could be developed by tumor cells during neoadjuvant therapy, possibly as a therapy escape mechanism. This must be validated in fundamental research projects. It is known that the silencing of *ASS1* leads to increased proliferation caused by facilitated pyrimidine synthesis [29]. Also, ASS1 pauses cell cycle progression following DNA damage. So, *ASS1* silencing promotes cancer mutagenesis, which facilitates the adaptability potential of cancer cells [30]. All ASS1-loss patients were non-responders to their neoadjuvant therapy, and all patients also suffered from vital lymph node metastases as an expression of a very high risk of recurrence. This underlines the need for a personalized treatment option with pegargiminase. This therapy option is also biomarker-supported via the detection of ASS1 expression in the tumor cells. ASS1 immunohistochemical staining is cost-effective and straightforward to implement in clinical practice.

However, our study has limitations. First, it is a retrospective analysis. While the percentage of ASS1 loss is mainly unaffected by the retrospective study design, especially the prognostic analyses could be altered by variables that were not collected in our data bank. Further prospective studies are needed to confirm ASS1 as a biomarker for patient prognosis. Second, variability in antibodies and cut-offs for defining ASS1 deficiency, as discussed earlier, complicates comparisons across studies. To address this, we selected an antibody with reliable results in esophageal adenocarcinoma using RNAScope. We emphasize the use of standardized cut-offs and consistent antibodies in future research. To enhance objectivity in the evaluation of immunohistochemical stainings, analyses could potentially be conducted digitally. As published by our group before, these analyses could be performed easily and are widely accessible, as QuPath v0.3.2 is an open-source software [31]. However, we conducted our analyses of this project following published clinical trials evaluating pegargiminase in ASS1-deficient patient cohorts. Despite these limitations, immunohistochemistry remains the gold standard for detecting ASS1 protein in the tumor cells, as supported by previous clinical studies investigating arginine deprivation therapy.

## 5. Conclusions

In summary, complete ASS1 loss was identified in 6.2% of patients with esophageal adenocarcinoma, with a further 6.2% showing low ASS1 expression (<10% tumor cells). Given the demonstrated efficacy of arginine deprivation therapy with pegargiminase in other cancer types and the identification of a significant patient subgroup with ASS1 deficiency in esophageal adenocarcinoma, clinical trials are warranted to evaluate the potential benefit of this therapy in this context. Moreover, incorporating immunohistochemical ASS1 staining into the diagnostic workflow for biopsies and tumor specimens could enhance the personalization of anti-cancer treatments, paving the way for more targeted and effective therapeutic strategies.

## Figures and Tables

**Figure 1 cancers-17-03624-f001:**
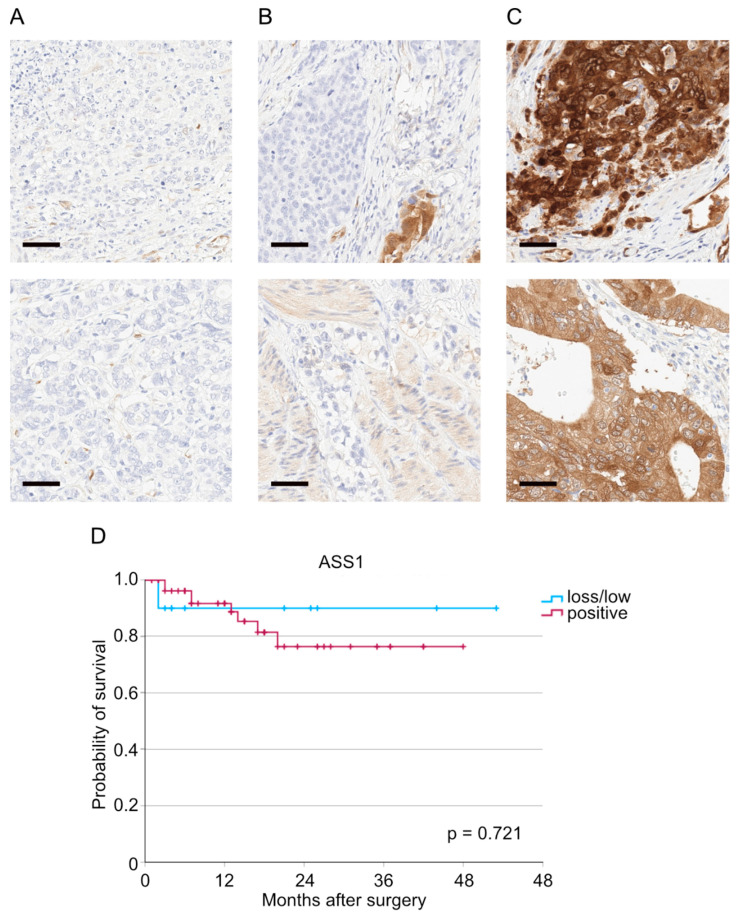
Representative pictures of tumors with (**A**) ASS1 loss, (**B**) low ASS1 expression, and (**C**) positive ASS1 expression. (**D**) Kaplan–Meier curve depending on ASS1 expression (*p* = 0.721, *n* (ASS1 loss/low) = 10, *n* (ASS1 positive) = 57). Sidebar: 50 µm.

**Table 1 cancers-17-03624-t001:** General patient characteristics of the entire patient cohort stratified by ASS1 expression status. (y)pN: pathological lymph node status (after neoadjuvant therapy), (y)pT: pathological tumor status (after neoadjuvant therapy), AEG: Siewert classification of the adenocarcinoma of the esophagogastric junction, CROSS: Chemoradiotherapy for Oesophageal Cancer followed by Surgery Study, FLOT: fluorouracil, leucovorin, oxaliplatin and docetaxel, L: lymph vessel infiltration, V: blood vessel infiltration. Bold print marks *p*-values below 0.05.

Characteristic		ASS1			
	Total	Loss	Low	Positive	*p*-Value
	*n* (%)	*n* (%)	*n* (%)	*n* (%)	
**No. of patients**	97 (100)	6 (100)	6 (100)	85 (100)	
**Sex**					0.600
Male	84 (86.6)	5 (83.3)	6 (100.0)	73 (85.9)	
Female	13 (13.4)	1 (16.7)	0 (0.0)	12 (14.1	
**Age**					**0.007**
≤65	38 (39.2)	2 (33.3)	6 (100.0)	30 (35.3)	
>65	59 (60.8)	4 (66.7)	0 (0.0)	55 (64.7)	
**AEG**					0.211
I	44 (45.4)	1 (16.7)	5 (83.3)	38 (44.7)	
II	51 (52.6)	5 (83.3)	1 (16.7)	45 (52.9)	
III	2 (2.1)	0 (0.0)	0 (0.0)	2 (2.4)	
**Perioperative/** **neoadjuvant therapy**					0.063
No	47 (48.5)	1 (16.7)	1 (16.7)	55 (64.7)	
Yes	50 (51.5)	5 (83.3)	5 (83.3)	45 (52.9)	
**Type of perioperative/ neoadjuvant therapy**					0.564
FLOT	35 (70.0)	5 (100.0)	3 (60.0)	27 (67.5)	
CROSS	13 (26.0)	0 (0.0)	2 (40.0)	11 (27.5)	
Other	2 (4.0)	0 (0.0)	0 (0.0)	2 (5.0)	
**Surgical approach**					0.581
thoracoabdominal esophagectomy	83 (85.6)	5 (83.3)	6 (100.0)	72 (84.7)	
transhiatal extended gastrectomy	14 (14.4)	1 (16.7)	0 (0.0)	13 (15.3)	
**Surgical technique**					0.687
Open	24 (24.7)	2 (33.3)	1 (16.7)	21 (24.7)	
Hybrid	54 (55.7)	4 (66.7)	3 (50.0)	47 (55.3)	
Total minimally invasive	19 (19.6)	0 (0.0)	2 (33.3)	17 (20.0)	
**Clavien-Dindo classification**					**0.018**
0	36 (37.1)	2 (33.3)	1 (16.7)	33 (38.3)	
I	3 (3.1)	0 (0.0)	2 (33.3)	1 (1.2)	
II	8 (8.2)	1 (16.7)	0 (0.0)	7 (8.2)	
IIIa	23 (23.7)	1 (16.7)	2 (33.3)	20 (23.5)	
IIIb	13 (13.4)	1 (16.7)	0 (0.0)	12 (14.1)	
IVa	8 (8.2)	1 (16.7)	0 (0.0)	7 (8.2)	
IVb	3 (3.1)	0 (0.0)	0 (0.0)	3 (3.5)	
V	3 (3.1)	0 (0.0)	1 (16.7)	2 (2.4)	
**Anastomotic leakage**					0.497
No	88 (90.7)	6 (100.0)	6 (100.0)	76 (89.4)	
Yes	9 (9.3)	0 (0.0)	0 (0.0)	9 (10.6)	
**(y)pT**					0.667
0	1 (1.0)	0 (0.0)	0 (0.0)	1 (1.2)	
1	19 (19.6)	0 (0.0)	0 (0.0)	19 (22.4)	
2	9 (9.3)	0 (0.0)	1 (16.7)	8 (9.4)	
3	61 (62.9)	5 (83.3)	5 (83.3)	51 (60.0)	
4	7 (7.2)	1 (16.7)	0 (0.0)	6 (7.1)	
**(y)pN**					0.348
0	31 (32.0)	0 (0.0)	2 (33.3)	29 (34.1)	
1	21 (21.6)	1 (16.7)	1 (16.7)	19 (22.4)	
2	20 (20.6)	1 (16.7)	1 (16.7)	18 (21.2)	
3	25 (25.8)	4 (66.6)	2 (33.3)	23 (22.4)	
**L**					0.297
0	24 (24.7)	0 (0.0)	1 (16.7)	23 (27.1)	
1	73 (75.3)	6 (100.0)	5 (83.3)	62 (72.9)	
**V**					0.057
0	50 (52.1)	1 (16.7)	1 (16.7)	48 (56.5)	
1	46 (46.9)	5 (83.3)	4 (66.6))	37 (43.5)	
Unknown	1 (1.0)	0 (0.0)	1 (16.7)	0 (0.0)	
**Grading**					0.893
1	1 (1.0)	0 (0.0)	0 (0.0)	1 1.2)	
2	21 (21.6)	0 (0.0)	0 (0.0)	21 (24.7)	
3	23 (23.7)	1 (16.7)	1 (16.7)	21 (24.7)	
Unknown/not. applicable	53 (54.6)	5 (83.3)	5 (83.3)	42 (49.4)	

**Table 2 cancers-17-03624-t002:** Results of univariable Cox regression analyses. (y)pN: pathological lymph node status (after neoadjuvant therapy), (y)pT: pathological tumor status (after neoadjuvant therapy), HR: hazard ratio, L: lymph vessel infiltration, V: blood vessel infiltration, vs.: versus. Bold print marks *p*-values below 0.05.

Characteristic	Borders	Hazard Ratio	95% Confidence Interval	*p*-Value
Sex	male vs. female	0.615	0.134–2.814	0.531
Age	≥65 vs. <65	0.455	0.144–1.438	0.180
Clavien-Dindo classification	≥IIIb vs. <IIIb	2.426	0.781–7.537	0.126
(y)pT	≥2 vs. <2	3.334	1.165–9.543	**0.025**
(y)pN	≥1 vs. 0	1.583	0.944–2.654	0.081
L	1 vs. 0	2.952	0.380–22.901	0.300
V	≥1 vs. 0	2.351	0.704–7.852	0.165
ASS1	positive vs. loss/low	0.882	0.337–2.313	0.799

**Table 3 cancers-17-03624-t003:** Results of multivariable Cox regression analyses. (y)pN: pathological lymph node status (after neoadjuvant therapy), (y)pT: pathological tumor status (after neoadjuvant therapy), HR: hazard ratio, V: blood vessel infiltration, vs.: versus. Bold print marks *p*-values below 0.05.

Characteristic	Borders	Hazard Ratio	95% Confidence Interval	*p*-Value
Age	≥65 vs. <65	0.331	0.094–1.167	0.085
Clavien-Dindo classification	≥IIIb vs. <IIIb	4.766	1.450–15.667	**0.010**
(y)pT	≥2 vs. <2	4.542	1.363–15.132	**0.014**
(y)pN	≥1 vs. 0	1.113	0.607–2.041	0.730
V	≥1 vs. 0	1.052	0.260–4.261	0.944

## Data Availability

The datasets generated and analyzed during the current study are available from the corresponding author upon reasonable request.

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
