# Peer review of "Loss of Argininosuccinate Synthetase-1 (ASS1) Occurs in Esophageal Adenocarcinoma and Represents a Promising Biomarker for Therapy with Pegargiminase"

_cancers, 2025, doi:10.3390/cancers17223624_

Round 1
Reviewer 1 Report
Comments and Suggestions for Authors
The aim of this manuscript is to investigate the expression level of ASS1 in biopsy samples from patients with esophageal adenocarcinoma, which may represent a potential target for arginine-deprivation therapy. Among the 97 samples analyzed, approximately 12.4% showed reduced or complete loss of ASS1 expression. This is a very interesting and potentially impactful study highlighting a novel therapeutic target in esophageal cancer. Several suggestions are provided below for consideration:
-
Please include additional representative images of ASS1 immunostaining, particularly examples showing low expression. At least two representative images for each expression category (normal, low, and loss) would strengthen the presentation.
-
Quantitative assessment of ASS1 staining should be provided to clearly distinguish among normal, low, and loss of expression groups. This would enhance the objectivity of the findings.
-
Including ASS1 staining in normal (non-malignant) esophageal tissue samples would greatly enhance the interpretation of the results. However, it is understandable if such samples are not currently available.
-
Validation of ASS1 expression using additional techniques such as ELISA, Western blot, or qPCR would further support the immunohistochemistry findings and strengthen the study’s conclusions.
-
The survival curve appears to suggest a potential beneficial outcome in patients with ASS1 loss within the first two years. Could the authors clarify whether there were any differences in treatment regimens between patients with and without ASS1 expression loss? This information would provide valuable clinical context.
Author Response
The aim of this manuscript is to investigate the expression level of ASS1 in biopsy samples from patients with esophageal adenocarcinoma, which may represent a potential target for arginine-deprivation therapy. Among the 97 samples analyzed, approximately 12.4% showed reduced or complete loss of ASS1 expression. This is a very interesting and potentially impactful study highlighting a novel therapeutic target in esophageal cancer. Several suggestions are provided below for consideration:
Comment 1:
Please include additional representative images of ASS1 immunostaining, particularly examples showing low expression. At least two representative images for each expression category (normal, low, and loss) would strengthen the presentation.
Response 1:
As requested, we added two images for each expression category in Figure 1.
Comment 2:
Quantitative assessment of ASS1 staining should be provided to clearly distinguish among normal, low, and loss of expression groups. This would enhance the objectivity of the findings.
Response 2:
The precise quantitative determination of therapeutically relevant biomarkers in tissue is an important aspect. This involves two steps: 1) checking the quality of the immunohistochemical antibody and 2) determining a cut-off value. We addressed point 1 by comparing the expression quality of our immunohistochemical antibody with ASS1 mRNA expression in tissue. To do this, we used an RNA in situ technique that visualizes ASS1 mRNA in tissue sections. The number of mRNA spots visible under a light microscope is then correlated with the protein expression result. This technique is an ideal gold standard in the establishment of antibodies. We can show here that the final immunohistochemical antibody used (we compared three different antibodies with the mRNA result) correlates with the mRNA range and provides reliable results. For point two, we referred to published data from clinical studies that used different cut-off values. We defined 10 % as a cut-off. To address this, we expanded the discussion of this matter in our discussion section.
In some clinical trials, a cut-off of 50 % was used; others used a cut-off of 5 %:
- Beddowes E, Spicer J, Chan PY, Khadeir R, Corbacho JG, Repana D, Steele JP, Schmid P, Szyszko T, Cook G, Diaz M, Feng X, Johnston A, Thomson J, Sheaff M, Wu BW, Bomalaski J, Pacey S, Szlosarek PW (2017) Phase 1 Dose-Escalation Study of Pegylated Arginine Deiminase, Cisplatin, and Pemetrexed in Patients With Argininosuccinate Synthetase 1-Deficient Thoracic Cancers. J Clin Oncol 35 (16):1778-1785. doi:10.1200/JCO.2016.71.3230
- Hall PE, Ready N, Johnston A, Bomalaski JS, Venhaus RR, Sheaff M, Krug L, Szlosarek PW (2020) Phase II Study of Arginine Deprivation Therapy With Pegargiminase in Patients With Relapsed Sensitive or Refractory Small-cell Lung Cancer. Clin Lung Cancer 21 (6):527-533. doi:10.1016/j.cllc.2020.07.012
- Szlosarek PW, Wimalasingham AG, Phillips MM, Hall PE, Chan PY, Conibear J, Lim L, Rashid S, Steele J, Wells P, Shiu CF, Kuo CL, Feng X, Johnston A, Bomalaski J, Ellis S, Grantham M, Sheaff M (2021) Phase 1, pharmacogenomic, dose-expansion study of pegargiminase plus pemetrexed and cisplatin in patients with ASS1-deficient non-squamous non-small cell lung cancer. Cancer Med 10 (19):6642-6652. doi:10.1002/cam4.4196
Comment 3:
Including ASS1 staining in normal (non-malignant) esophageal tissue samples would greatly enhance the interpretation of the results. However, it is understandable if such samples are not currently available.
Response 3:
To address this point, we examined thirteen normal mucous membranes for their ASS1 expression (squamous epithelial esophageal mucosa: n = 10, Barrett's mucosa: n = 3). ASS1 is expressed in the basal keratinocytes of normal squamous epithelial esophageal mucosa. The remaining maturing squamous epithelia are ASS1-negative. The foveolar epithelia of the Barrett's mucosa show a low ASS1 expression. The expression level is weaker than in tubular adenocarcinomas. Additionally, we screened endothelial cells as internal positive controls of our ASS1 stainings. These cells show a strong and well-documented physiological ASS1 expression. We added this information to our manuscript.
Comment 4:
Validation of ASS1 expression using additional techniques such as ELISA, Western blot, or qPCR would further support the immunohistochemistry findings and strengthen the study’s conclusions.
Response 4:
Important aspect. We performed the validation using a tissue-based mRNA in situ measurement as described in Response 2. We performed RNAScope stainings and compared the results with immunohistochemical stainings of three different antibodies in an initial step. Here, we chose the antibody, which showed the closest correlation with the RNAScope results (Suppl. Fig. 1).
Comment 5:
The survival curve appears to suggest a potential beneficial outcome in patients with ASS1 loss within the first two years. Could the authors clarify whether there were any differences in treatment regimens between patients with and without ASS1 expression loss? This information would provide valuable clinical context.
Response 5:
We added the requested information regarding the perioperative/neoadjuvant therapy regimen to Table 1. Therapeutic regimens did not differ between the various ASS1 expression levels.
Reviewer 2 Report
Comments and Suggestions for Authors
This is a clearly written study that addresses clinically important question regarding the prevalence of ASS1 loss in esophageal adenocarcinoma.
I have the following comments:
-
The finding that all patients with ASS1 loss who received neoadjuvant therapy were non-responders is striking. Could the authors please confirm if the "minor response" mentioned in the text corresponds to a specific tumor regression grade (e.g., TRG 2-3)?
-
Suggesting ASS1 loss/low is more frequent in the neoadjuvant-treated group is intriguing. To help interpret this, could the authors clarify if the 47 patients who didn't receive neoadjuvant therapy were all stage I radiologically or there was another reason for upfront surgery?
-
The link between younger age and ASS1 loss/low is interesting. While perhaps beyond the study's scope, is there any known biological or etiological link?
- Authors should describe the location of tumours (uper, mid, lower esophagus or GEJ).
-
The manuscript states patients underwent "curative-intent surgery". The authors should specify the surgical approach (Ivor Lewis, total esophagectomy, transhiatal), open or MIS and the extent of lymphadenectomy (2-field, 3-field).
- The authors should also add the frequency of adjuvant therapy. Did the patients received chemotherapy, radiotherapy, immunotherapy? These data should also be provided for neoadjuvant therapy.
-
The study includes survival analyses. However, postoperative morbidity and mortality data (e.g., Clavien-Dindo classification) are not presented. Given that postoperative complications (eg. AL) are strongly associated with long-term survival, could the authors include this data and analyze its potential as a confounding variable in the survival analysis?
-
The Methods section states that univariable and multivariable Cox regression analyses were planned. Please provide the hazard ratio tables for the univariable and multivariable analyses, including ASS1 status and other key prognostic variables.
-
Please revise grammar and syntax throughout the text.
- Format references according to the journal's style.
Author Response
This is a clearly written study that addresses clinically important question regarding the prevalence of ASS1 loss in esophageal adenocarcinoma.
I have the following comments:
Comment 1:
The finding that all patients with ASS1 loss who received neoadjuvant therapy were non-responders is striking. Could the authors please confirm if the "minor response" mentioned in the text corresponds to a specific tumor regression grade (e.g., TRG 2-3)?
Response 1:
We defined Minor response as grade I and II of the Cologne regression grade system, which translates to ³ 10 % vital residual tumor cells following neoadjuvant therapy as published before (Schneider PM, Baldus SE, Metzger R, Kocher M, Bongartz R, Bollschweiler E, Schaefer H, Thiele J, Dienes HP, Mueller RP et al. Histomorphologic tumor regression and lymph node metastases determine prognosis following neoadjuvant radiochemotherapy for esophageal cancer: implications for response classification. Ann Surg. 2005;242(5):684-92).
We added this crucial information to our manuscript.
Comment 2:
Suggesting ASS1 loss/low is more frequent in the neoadjuvant-treated group is intriguing. To help interpret this, could the authors clarify if the 47 patients who didn't receive neoadjuvant therapy were all stage I radiologically or there was another reason for upfront surgery?
Response 2:
36.2 % (n = 17) of those patients who underwent primary surgery were diagnosed with disease stage I initially. In contrast, 30 patients (63.8%) either received a recommendation for primary surgery from our interdisciplinary tumor board conference due to multimorbidity or advanced age, or declined perioperative therapy and opted for primary resection. We added this information to our manuscript.
Comment 3:
The link between younger age and ASS1 loss/low is interesting. While perhaps beyond the study's scope, is there any known biological or etiological link?
Response 3:
We agree that this is indeed an interesting observation. Therefore, we added the following paragraph to our discussion section:
Interestingly, we could describe that patients with ASS1 loss or low ASS1 expression were significantly younger compared to patients with positive ASS1 expression in our cohort (p = 0.007). Previous studies have suggested that esophageal adenocarcinomas may be biologically more aggressive in younger patients. In a retrospective cohort study of patients with esophageal adenocarcinoma, age younger than 45 years was associated with more advanced tumor and lymph node status. Furthermore, these patients showed a significantly worse disease-specific survival compared to older age groups (27). On the other hand, ASS1 deficiency could be linked to higher cell proliferation rates in in vitro experiments (28, 29), while an ASS1 knockout in mouse embryonic stem cells led to decreased apoptosis (28). We propose that ASS1 deficiency contributes to a more aggressive tumor biology, which may explain its higher prevalence in younger patients.
Comment 4:
Authors should describe the location of tumours (upper, mid, lower esophagus or GEJ).
Response 4:
We added the information on the location of the tumours in Table 1. No significant differences in tumor location could be observed regarding the ASS1 expression level.
Comment 5:
The manuscript states patients underwent "curative-intent surgery". The authors should specify the surgical approach (Ivor Lewis, total esophagectomy, transhiatal), open or MIS and the extent of lymphadenectomy (2-field, 3-field).
Response 5:
As requested, the mentioned information was added to Table 1 and described in our results section.
Comment 6:
The authors should also add the frequency of adjuvant therapy. Did the patients received chemotherapy, radiotherapy, immunotherapy? These data should also be provided for neoadjuvant therapy.
Response 6:
We added information about the perioperative/neoadjuvant therapy regimen to Table 1. These therapeutic regimens did not differ between the various ASS1 expression levels. However, as a supraregional center of excellence for upper gastrointestinal surgery, patients are referred to us nationwide. Since adjuvant therapy is mainly provided by local outpatient departments, the quality of our data on adjuvant treatment among the included patients is not sufficiently reliable for publication.
Comment 7:
The study includes survival analyses. However, postoperative morbidity and mortality data (e.g., Clavien-Dindo classification) are not presented. Given that postoperative complications (eg. AL) are strongly associated with long-term survival, could the authors include this data and analyze its potential as a confounding variable in the survival analysis?
Response 7:
We added the Clavien-Dindo classification as well as the occurrence of an anastomotic leakage to Table 1. Furthermore, we included the Clavien-Dindo classification as an additional variable in our Cox regression analyses. Here, a higher Clavien-Dindo classification was a significant factor for worse patient survival (HR = 4.766, 95 % confidence interval = 1.450-15.677, p = 0.010).
To minimize this confounding factor, we excluded patients with an overall survival of less than 30 days from the survival analyses in our initial study design, as early postoperative mortality is primarily related to complications following resection rather than oncological outcome.
Comment 8:
The Methods section states that univariable and multivariable Cox regression analyses were planned. Please provide the hazard ratio tables for the univariable and multivariable analyses, including ASS1 status and other key prognostic variables.
Response 8:
As suggested by the reviewer, we added the results of the univariable and multivariable Cox regression analyses as Tables 2 and 3.
Comment 9:
Please revise grammar and syntax throughout the text.
Response 9:
We revised our manuscript thoroughly to improve grammar and syntax.
Comment 10:
Format references according to the journal's style.
Response 10:
We revised the references according to the editors’ comments.
Round 2
Reviewer 1 Report
Comments and Suggestions for Authors
The author response all the comments with modification of the manuscript.
Reviewer 2 Report
Comments and Suggestions for Authors
Authors addressed my comments.